# Prevalence and impact of combined vision and hearing (dual sensory) impairment: A scoping review

Tess Bright[1]*, Jacqueline Ramke[2,3], Justine H. Zhang[2,4], Gatera Fiston Kitema[5], Sare Safi[6], Shaffi Mdala[7], Miho Yoshizaki[2], Christopher G. Brennan-Jones[8,9,10], Islay Mactaggart[2,11], Iris Gordon[2], Bonnielin K. Swenor[12,13,14], Matthew J. Burton[2,15], Jennifer R. Evans[2,16]

1 Indigenous Health Equity Unit, Centre for Health Equity, University of Melbourne, Melbourne, Australia, 2 International Centre for Eye Health, London School of Hygiene & Tropical Medicine, London, United Kingdom, 3 School of Optometry and Vision Science, University of Auckland, Auckland, New Zealand, 4 Manchester Royal Eye Hospital, Manchester, United Kingdom, 5 Ophthalmology Department, School of Health Sciences, University of Rwanda, Kigali, Rwanda, 6 Ophthalmic Epidemiology Research Center, Research Institute for Ophthalmology and Vision Science, Shahid Beheshti University of Medical Sciences, Tehran, Iran, 7 Ophthalmology Department, Queen Elizabeth Central Hospital, Blantyre, Malawi, 8 Ear Health Group, Telethon Kids Institute, The University of Western Australia, Perth, Australia, 9 Faculty of Health Sciences, Curtin University, Perth, Australia, 10 Department of Audiology, Perth Children's Hospital, Nedlands, Western Australia, Australia, 11 International Centre for Evidence in Disability, London School of Hygiene & Tropical Medicine, London, United Kingdom, 12 The Johns Hopkins Disability Health Research Center, Johns Hopkins University, Baltimore, Maryland, United States of America, 13 The Johns Hopkins School of Nursing, Johns Hopkins University, Baltimore, Maryland, United States of America, 14 Department of Epidemiology, Johns Hopkins School of Medicine, Johns Hopkins Bloomberg School of Public Health, The Wilmer Eye Institute, Johns Hopkins University, Baltimore, Maryland, United States of America, 15 National Institute for Health Research Biomedical Research Centre for Ophthalmology at Moorfields Eye Hospital NHS Foundation Trust and UCL Institute of Ophthalmology, London, United Kingdom, 16 Centre for Public Health, Queens University Belfast, Belfast, United Kingdom

* brightt@unimelb.edu.au

**Data Availability Statement:** Data is available on request from International Centre for Evidence in Disability via email address: disabilitycentre@lshtm.ac.uk.

## Abstract

Hearing and vision impairments are common globally. They are often considered separately in research, and in planning and delivering services. However, they can occur concurrently, termed dual sensory impairment (DSI). The prevalence and impact of hearing and vision impairment have been well-examined, but there has been much less consideration of DSI. The aim of this scoping review was to determine the nature and extent of the evidence on prevalence and impact of DSI. Three databases were searched: MEDLINE, Embase and Global Health (April 2022). We included primary studies and systematic reviews reporting the prevalence or impact of DSI. No limits were placed on age, publication dates, or country. Only studies where the full text was available in English were included. Two reviewers independently screened titles, abstract, full texts. Data were charted by two reviewers independently using a pre-piloted form. The review identified 183 reports of 153 unique primary studies and 14 review articles. Most evidence came from high-income countries (86% of reports). Prevalence varied across reports, as did age groups of participants and definitions used. The prevalence of DSI increased with age. Impact was examined across three broad groups of outcomes—psychosocial, participation, and physical health. There was a strong trend towards poorer outcomes for people with DSI across all categories compared to

**Funding:** MJB is supported by the Wellcome Trust (207472/Z/17/Z). JR's appointment at the University of Auckland is funded by the Buchanan Charitable Foundation, New Zealand. TB is funded by Christian Blind Mission (CBM International). The Lancet Global Health Commission on Global Eye Health is supported by The Queen Elizabeth Diamond Jubilee Trust, Moorfields Eye Charity [grant number GR001061], NIHR Moorfields Biomedical Research Centre, Wellcome Trust, Sightsavers, The Fred Hollows Foundation, The SEVA Foundation, British Council for the Prevention of Blindness and Christian Blind Mission. CGBJ is supported by an NHMRC Fellowship (GNT 1142897) and a WA Future Health Research and Innovation Fund Fellowship. The funders had no role in study design, data collection and analysis, decision to publish, or preparation of the manuscript.

**Competing interests:** The authors have declared that no competing interests exist.

people with one or neither impairment, including activities of daily living (worse for people with DSI in 78% of reports) and depression (68%). This scoping review highlights that DSI is a relatively common condition with substantial impact, particularly among older adults. There is a gap in evidence from low and middle-income countries. There is a pressing need for a consensus position on the definition(s) of DSI and standardisation of reporting age groups to enable reliable estimates to be ascertained and compared and responsive services developed.

## Introduction

Hearing and vision impairments are common globally [1, 2]. These two sensory impairments are often considered separately in research as well as in planning and delivering services. However, they can occur concurrently, which is commonly termed dual sensory impairment (DSI), dual sensory loss, combined/concurrent vision and hearing impairment, deafblindness or multi-sensory impairment [3]. In this review we use the term DSI. The global prevalence of DSI has not been fully examined, however it is thought to be more common in older people, as the prevalence of both vision and hearing impairment increase with age [4]. A report by the World Federation for the Deafblind estimated that 0.2% of the global population are living with deafblindness, which excludes those with milder forms of DSI who may still experience barriers to participation. In the context of the global ageing population, the prevalence of DSI is expected to increase [5]. This has implications for the wellbeing and quality of life for people with, or at risk of, DSI, as well as for delivering effective health services to maximise their health and functional ability.

People can develop DSI at different points in their life and the impact of DSI may depend on when it occurs in their life course. Some people have DSI from birth (congenital), some develop DSI during early childhood or have one impairment from childhood and develop the other later (acquired), but most people with DSI acquire vision and hearing impairment later in life (age-related deafblindness) [6]. There are internationally agreed World Health Organization (WHO) definitions of hearing and vision impairment separately, and some agreement on within the deafblindness field through the Nordic definition of deafblindness for rehabilitation and service delivery [7]. However, there is currently no consensus on how to measure and define DSI, particularly in terms of the severity of the underlying impairments when they co-exist, and whether a definition should rely on behaviously measurable observations (e.g. acuity, audiogram) or functional impairment (subjectively reported).

The independent impact of either vision loss or hearing loss on quality of life, wellbeing, participation, and mental health has been well explored [8–16], along with the benefits of correcting these impairments [13, 17, 18]. Less is known about the impact of DSI on people's lives, though a small number of literature reviews and systematic reviews have examined quality of life [19], mental health [20], independence [21] and the range of impacts [22] among older people with DSI, as well as vulnerability [6], and participation [5, 23]. Many studies and reviews focus on people who develop DSI later in life, high-income contexts and report only one specific outcome. There is a need to examine the evidence on the prevalence of DSI more broadly, and to explore the potential wide-reaching and multitude of effects on people's lives. This type of evidence can inform interventions to improve health and quality of life for people with DSI. Furthermore, there is a need to examine how DSI has been defined in the literature and work towards a standardised definition.

We undertook a scoping review to identify and map the available evidence on these themes, and anticipated heterogenous evidence [24, 25]. The aim of this scoping review was to determine the nature and extent of the evidence on DSI relating to:

1. The definitions of DSI used in the literature;

2. The prevalence of DSI globally and across regions for all age groups; and

3. The impact of DSI on people's lives (e.g. quality of life, mental health, mortality).

## Methods and analysis

### Ethics statement

As this study only included published data, ethics approval was not sought.

This scoping review was undertaken as part of the *Lancet Global Health* Commission on Global Eye Health [26]. The methods and results are reported according to the relevant items of the Preferred Reporting Items for Systematic reviews and Meta-Analyses extension for Scoping Reviews (PRISMA-ScR) checklist [24, 27]. This protocol was registered on Open Science Framework (OSF) (DOI: 10.17605/OSF.IO/MGYFV). We decided to undertake a scoping review rather than a systematic review in order to identify and map the available evidence on these themes, and expected the results to be heterogenous [24, 25].

### Eligibility criteria

**Study type.** We included primary studies and systematic reviews. No time limit was placed on publication dates. We excluded studies not reported in English, editorials, interviews, case reports, and comments and studies where the full text was not available. We excluded reviews that were not systematic (e.g. narrative reviews that did not demonstrate a systematic search in the methodology). We included systematic reviews to understand which aspects of DSI have received the most attention by researchers synthesising the evidence.

**Participants.** Only studies involving human participants were included. There were no age restrictions. All types of combined hearing and vision impairment were included, whether measured via self-report, clinical tools, or through registries. Studies that focussed on syndromes (e.g. Usher's syndrome) were included if results for people with DSI were disaggregated in the study.

**We excluded.** Studies that focussed on causes of DSI among a restricted population subgroup (e.g. pre-term infants) unless they reported impact outcomes, because findings in these studies would not be applicable to people with DSI generally; studies that only considered hearing and vision impairment separately; and studies that reported on the prevalence (not impact) of DSI amongst children from schools for the deaf or blind, as these would not be representative of DSI prevalence in the general population.

**Context.** No limits were placed on country of study. We excluded studies that focussed on service provision (e.g. screening techniques) for people with DSI because we aimed to scope only studies reporting prevalence and/or impact.

**Outcomes.** Studies that reported the prevalence or the impact of DSI, were included. No restrictions were placed on types of impact outcome measures. We anticipated outcomes related to health and well-being such as mental health, mortality, quality of life, participation, falls, trauma or education [6, 19–21, 23, 28]. We considered impact as a consequence of DSI, rather than a cause.

## Search strategy and information sources

Three databases were searched from inception to April 2022 (MEDLINE, Embase and Global Health) using rigorous search strategies that were developed and run by an experienced Cochrane Information Specialist (IG). The search strategy can be found in the supplementary material. Reference lists of included articles were examined to identify any further eligible articles (snowballing).

## Study selection

Two reviewers independently screened titles and abstracts of identified studies using Covidence systematic review management software (Veritas Health Innovation, Melbourne, Australia. Available at www.covidence.org). We selected all reports for full text screening where DSI (or equivalent terms) were mentioned in the title or abstract of the report. Full text articles were obtained and reviewed independently by two reviewers for relevance. Any discrepancies between the reviewers were solved by a discussion with a third reviewer. A PRISMA flow chart was compiled to display the study selection process (Fig 1).

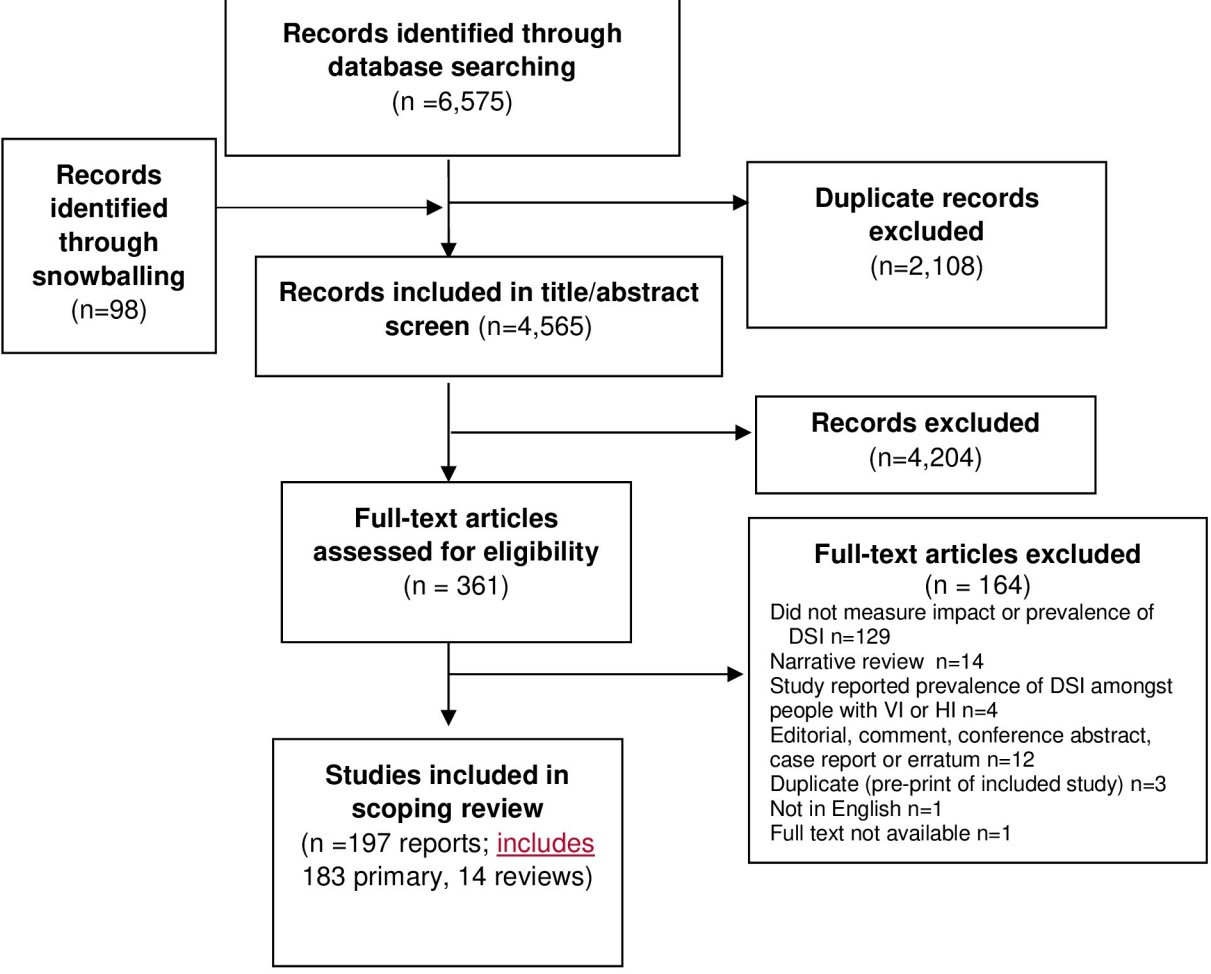

**Fig 1. PRISMA flow chart.** DSI: Dual sensory impairment, HI: Hearing impairment, VI: Vision impairment].

## Data charting and synthesis of results

A data extraction form was developed in Google Forms. This form was piloted on three studies by three reviewers, and amendments made as necessary. Data were extracted by two reviewers independently to ensure accuracy. Any discrepancies between the reviewers were solved by a discussion with a third reviewer.

Items for data extraction included:

All studies:

- Publication characteristics: title, year of publication, countries of study and source of funding, World Bank country income group (at time of publication)

  For primary studies:

- Characteristics of study: year(s) of data collection, design, sample size (total and DSI), study setting, recruitment characteristics, age group of the study population, method of impairment measurement and definition of individual and dual impairments

- Outcome measures: prevalence (including confidence interval if reported) and impact outcomes

- The type of syndrome: for primary studies that focussed on syndromes

  For systematic reviews:

- Number of included studies

- Types of included studies

- Scope of the research (e.g. research questions)

- Review conclusions

Extracted data were coded in Excel and imported into Stata v15.0 for descriptive analysis. Where confidence intervals were not reported around prevalence estimates, these were calculated in Stata using the *cii proportion* command. Prevalence data were graphed using forest plots in Stata. Data were summarised using a narrative synthesis. The study findings were grouped according to the outcomes measured with three key themes–definitions of DSI, prevalence of DSI and impact outcomes. The types of impact outcomes were mapped in terms of what has been measured, and the resulting impact. Outcomes were classified into broad themes to allow summaries to be made. Each outcome type was classified in terms of whether the outcome among people with DSI was "worse", "better" or "the same" compared to a relevant comparison group. These outcomes were judged based on statistical significance (e.g. p values, or confidence intervals). If studies measured multiple outcomes (e.g., mental health and quality of life), studies were classified as "worse", "the same", or "better" if all outcomes showed the same relationship. Impact data summary outcomes were summarised in bar charts created in Excel. Studies were classified as "varied" if results for a study were worse in one outcome of interest or and the same or better in another. We planned to present impact results overall as well as for each Global Burden of Disease super-region, however due to insufficient data from some regions of the world this was not done.

## Patient and public involvement statement

This scoping review was developed with input from the Commissioners of the Lancet Global Health Commission on Global Eye Health, which includes people with lived experience of

vision impairment, policy makers, academics, clinicians, government eye health programme leaders and advocacy specialists.

## Results

### Description of included studies

Overall, 6,575 articles were identified through database searching, and an additional 98 through snowballing. After duplicates were removed, titles and abstracts of 4,565 articles were screened and 361 selected for full text review. Following this, 197 reports were included based on the inclusion/exclusion criteria (Fig 1) [29].

Of the included reports, 14 were reviews and the 183 primary reports came from 153 unique primary studies (i.e. multiple publications per study). As different outcomes were included in individual reports we present results by report, indicating when findings come from the same study as appropriate. Table 1 summarises the characteristics of included reports of primary studies. Regionally, approximately two-thirds of reports came from either North America (n = 72, 39%) or Western Europe (n = 49, 27%); none were from sub-Saharan Africa. Studies were conducted in 35 countries, with nearly 9 in 10 reports coming from high-income countries (n = 158, 86%). More than three-quarters of reports were from studies conducted since 2010 (n = 142, 78%).

Studies from over half of reports used cross-sectional designs (n = 106; 58%). Cohort studies were also common– 31% of reports were from prospective and 6% were retrospective cohorts. A small number of qualitative reports were included (n = 5, 3%). Most reports recruited participants from the population (n = 123, 67%) or were clinic-based (n = 20, 11%). Other locations included care homes and registers of people with disability. The majority of reports (n = 122, 67%) included older adults only (≥40 years); very few reports included only children (n = 4, 2%).

Many reports included more than one outcome—66% (n = 121) presented prevalence of DSI and 85% (n = 156) presented impact outcomes, including psychosocial health, participation, and physical health. Within each of these broad categories a range of outcomes and measures were used, and these are discussed further below.

### Definition and measurement of DSI

A range of methods were used to measure, and thus define DSI across reports. Hearing impairment (HI) was most commonly measured via self- or proxy-report alone (n = 95, 52%), followed by pure-tone audiometry (n = 58, 32% and a further four studies used both pure-tone audiometry and self-report) (Table 2). Vision impairment (VI) was most commonly assessed via self- or proxy-reported measures (alone) (n = 88, 48%), or a visual acuity (VA) chart (n = 70, 38%, a further two also used self-report, in combination with VA charts). Most reports defined DSI as a combination of both VI and HI according to the definitions of each single impairment.

Self-reported measures of each of HI and VI included a range of different tools/questions but commonly used a single question with a response on a Likert scale, or with binary or categorical answer (Table 2). Among the reports that used visual acuity charts to assess VI, there were more than 19 different definitions, with variation in the chart used, whether better/worse or both eyes and the visual acuity threshold (Table 2). The most common definition was bilateral / better eye VA <6/12 (n = 24, 33%) which equates to mild VI in ICD11 [30]. There were more than 23 different definitions of HI across reports, with variation by decibel cut-off, frequencies included, and focus on the better or worse ear (Table 2). The most common

**Table 1.  Characteristics of 183 reports of 153 primary studies reporting dual sensory impairment prevalence or impact.**

| Characteristic | | n | % (n/183) |
|---|---|---|---|
| Region* | HIC—North America | 72 | 39.3 |
| | HIC—Australasia | 14 | 7.7 |
| | HIC—Asia Pacific | 12 | 6.6 |
| | HIC—Western Europe | 49 | 26.8 |
| | Southeast Asia, East Asia, Oceania | 18 | 9.8 |
| | North Africa and Middle East | 2 | 1.1 |
| | South Asia | 6 | 3.3 |
| | Latin America and the Caribbean | 2 | 1.1 |
| | Multiple | 7 | 3.8 |
| | Not specified | 1 | 0.5 |
| World Bank Country income group** | Low | 1 | 0.5 |
| | Lower middle | 6 | 3.3 |
| | Upper middle | 17 | 9.3 |
| | High | 158 | 86.3 |
| | Not specified | 1 | 0.5 |
| Decade of publication | 1980 | 1 | 0.5 |
| | 1990 | 9 | 4.9 |
| | 2000 | 31 | 16.9 |
| | 2010 | 91 | 49.7 |
| | 2020 (to search date) | 51 | 27.9 |
| Study design | Cross sectional | 106 | 57.9 |
| | Prospective cohort | 56 | 30.6 |
| | Retrospective cohort study | 11 | 6.0 |
| | Qualitative | 5 | 2.7 |
| | Case control | 2 | 1.1 |
| | Secondary analysis | 1 | 0.5 |
| | Chart review | 1 | 0.5 |
| | Case series | 1 | 0.5 |
| Study setting | Population | 123 | 67.2 |
| | Clinic | 20 | 10.9 |
| | Register of people with disability | 14 | 7.7 |
| | Other | 14 | 7.7 |
| | Care home | 11 | 6.0 |
| | Not specified | 1 | 0.5 |
| Age group (years)^^ | All ages | 6 | 3.3 |
| | Only children (<18) | 4 | 2.2 |
| | Adults ≥18 | 28 | 15.3 |
| | Older adults ≥40*** | 122 | 66.7 |
| | Older adults ≥70 | 22 | 12.0 |
| | Unknown | 1 | 0.5 |

(*Continued*)

**Table 1.** (Continued)

| Characteristic | | | n | % (n/183) |
|---|---|---|---|---|
| Outcomes | | Prevalence | 121 | 66.1 |
| | | Impact (any)^ | 156 | 85.2 |
| | | Impact—Psychosocial | 93 | 50.8 |
| | | Impact—Participation | 55 | 30.1 |
| | | Impact—Physical | 56 | 30.6 |

*GBD Super-region, with high-income country (HIC) super-region disaggregated into regions;

**At time of publication;

***includes diverse set of age groups;

^includes psychosocial, participation, and physical;

^^age group categories mutually exclusive

definition was a pure tone average (PTA) across frequencies 500, 1000, 2000, 4000Hz of >25dB in the better ear (n = 18 reports).

Given the range of tools and approaches used to measure VI and HI individually, the resulting definitions of DSI also varied substantially across studies—we identified 75 different definitions of DSI across these 183 reports. One of our included reviews reported the range of terminology used across its included primary studies—in addition to DSI, other terms were combined functional sensory impairment, dual sensory loss, double disability, and concurrent vision and hearing impairment [27]. Another included review focused on deafblindness and reported that half of the 29 included studies did not define the condition [28]. The large variation in definitions made synthesis challenging, and this is discussed in the following sections.

## Prevalence of DSI

Of 121 reports that measured prevalence of DSI, four did so for all ages, one in children (<18 years), fifteen in adults ≥18 years, and 101 amongst older adults aged ≥40 years. Overall, at least 15 different age cut-offs were used, making comparison difficult (Figs 2 and 3). This was exacerbated by the wide range of clinical test methods and thresholds for HI and VI described above. However, overall, there was a trend for increasing prevalence of DSI with increasing age. Studies of people aged 18+ had a median prevalence of DSI in the order of 3% and studies of people aged ≥65 years had a median prevalence over double that of approximately 7%. In one study in people over the age of 95 years, over 1 in every 3 people had DSI.

**All-age prevalence.** Of the four studies including people of all ages, three were population-based. The prevalence varied substantially across these studies. For the three population-based studies [31–33], a register-based study in Denmark [31] had the lowest prevalence of DSI of 0.003% (sample n = 190, VA <6/60 [eye not specified], PTA: 3 frequency average (3FA) ≥80 dB [ear not specified]) while a study in India had the highest prevalence of 1.9% (n = 3,574, VA <6/12 better eye, PTA ≥35dB children, ≥41dB adults better ear) [32]. The third population-based study, conducted in Oman had a prevalence of 0.25% (n = 11,400, VA<6/120 better eye; PTA: 3FA ≥36dB better ear) [33]. In the non-population-based study—which may not be representative—the prevalence was 0.015% in Canada (n = 564; chart review; VA ≤6/18 better eye; PTA ≥25dB PTA in better ear) [34].

**Prevalence amongst children.** The one study that included only children was a school-based study conducted in Sweden—among all school students in Sweden it reported a prevalence of DSI of 0.3% (n = 7,793; self-reported) [35].

**Table 2. Measurement and definition of hearing and vision impairments among 183 studies reporting dual sensory impairment.**

| | Hearing impairment | n | % | | Vision impairment | n | % |
|---|---|---|---|---|---|---|---|
| Measurement of hearing impairment (n = 183) | Self- or proxy-reported | 95 | 51.9 | Measurement of vision impairment (n = 183) | Self- or proxy-reported | 88 | 48.1 |
| | Pure tone audiometry | 58 | 31.7 | | Visual acuity chart | 70 | 38.3 |
| | Whisper voice test | 5 | 2.7 | | Self-report and visual acuity chart | 2 | 1.1 |
| | Self-report and pure tone audiometry | 4 | 2.2 | | Other | 11 | 6.0 |
| | Other | 10 | 5.5 | | Not specified | 12 | 6.6 |
| | Not specified | 11 | 6.0 | | | | |
| Self- or proxy-reported measures of hearing impairment (n = 99) | Single question Likert | 32 | 32.3 | Self- or proxy reported measures of vision impairment (n = 90) | Single question Likert | 30 | 33.3 |
| | Single question categorical | 31 | 31.3 | | Single question categorical | 24 | 26.7 |
| | Single question binary (e.g. yes/no) | 20 | 20.2 | | Single question binary (e.g. yes/no) | 22 | 24.4 |
| | Multiple questions | 15 | 15.2 | | Multiple questions | 11 | 12.2 |
| | Not specified / unclear | 1 | 1.0 | | Not specified / unclear | 2 | 2.2 |
| | | | | | Other | 1 | 1.1 |
| Definition of hearing impairment by pure tone audiometry (n = 62) | *PTA 500, 1000, 2000, 4000 Hz, better ear* | | | Definition of vision impairment by visual acuity measurement Snellen (LogMAR) (n = 72) | *Distance, better eye** | | |
| | ≥21dB | 1 | 1.7 | | <3/60 (1.30) | 1 | 1.4 |
| | ≥25dB | 4 | 6.9 | | ≤6/60 (1.00) | 2 | 2.8 |
| | >25dB | 18 | 31.0 | | <6/18 (0.50) | 9 | 12.5 |
| | ≥26dB | 5 | 8.6 | | ≤6/18 (0.50) | 1 | 1.4 |
| | ≥30dB | 1 | 1.7 | | <6/15 (0.40) | 3 | 4.2 |
| | ≥35dB | 3 | 5.2 | | ≤6/15 (0.40) | 2 | 2.8 |
| | ≥40dB | 4 | 6.9 | | <6/12 (0.30) | 24 | 33.3 |
| | >40dB | 8 | 13.8 | | ≤6/12 (0.30) | 5 | 6.9 |
| | >41dB adults, >35dB children | 1 | 1.7 | | <6/7.5 (0.10) | 3 | 4.2 |
| | ≥41dB | 2 | 3.4 | | *Distance, worse eye* | | |
| | >70dB | 2 | 3.4 | | <6/18 (0.50) | 1 | 1.4 |
| | Based on PTA but definition not clear | 1 | 1.7 | | <6/12 (0.30) | 2 | 2.8 |
| | *Three frequency average (500, 1000, 2000Hz)* | | | | *Distance, eye not specified* | | |
| | ≥26dB in better ear | 1 | 1.7 | | <1/60 (1.80) | 1 | 1.4 |
| | >30dB worse ear | 1 | 1.7 | | <6/60 (1.00) | 2 | 2.8 |
| | >40dB better ear | 1 | 1.7 | | <6/15 (0.40) | 1 | 1.4 |
| | ≥80dB ear not specified | 2 | 3.4 | | <6/12 (0.30) | 4 | 5.6 |
| | *Based on single frequency threshold in better ear* | | | | <6/9 (0.20) | 1 | 1.4 |
| | >30dB at 1k in better ear | 1 | 1.7 | | ≤6/9 (0.20) | 1 | 1.4 |
| | >40dB at 1kHz in better ear | 1 | 1.7 | | *Near, better eye* | | |
| | ≥40dB at 2kHz in better ear | 1 | 1.7 | | ≤20/70 (0.5) | 2 | 2.8 |
| | >40dB at 1 or/and 2kHz in better ear | 1 | 1.7 | | *Other* | 7 | 9.7 |
| | *Other* | 3 | 5.2 | | | | |

*10 of these measured binocular vision; which equates approximately to better eye vision; dB: decibels PTA: pure tone average

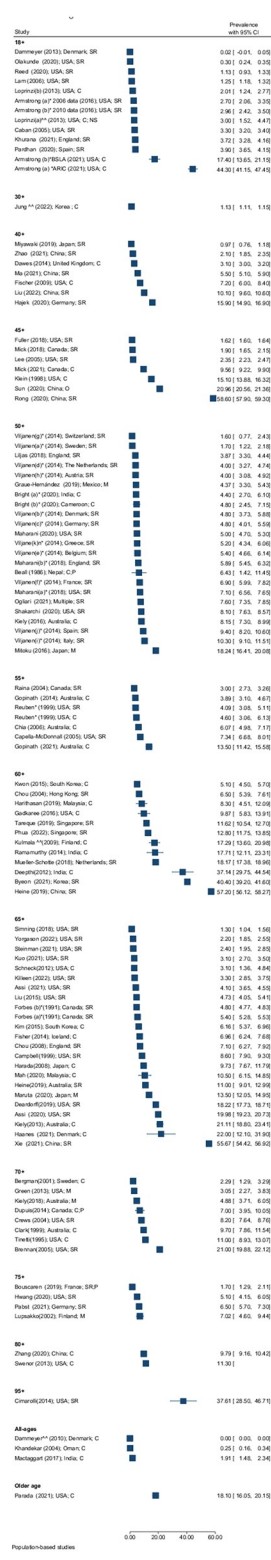

**Fig 2. Prevalence (95%CI) of DSI by age, region, study setting, and measurement type in population-based studies.** (SR = self-report; C = clinical (PTA or VA); M = mixed (SR and C); O = other)] ^^ measured DSI in a subpopulation, * same study, multiple countries, or multiple prevalence estimates reported.

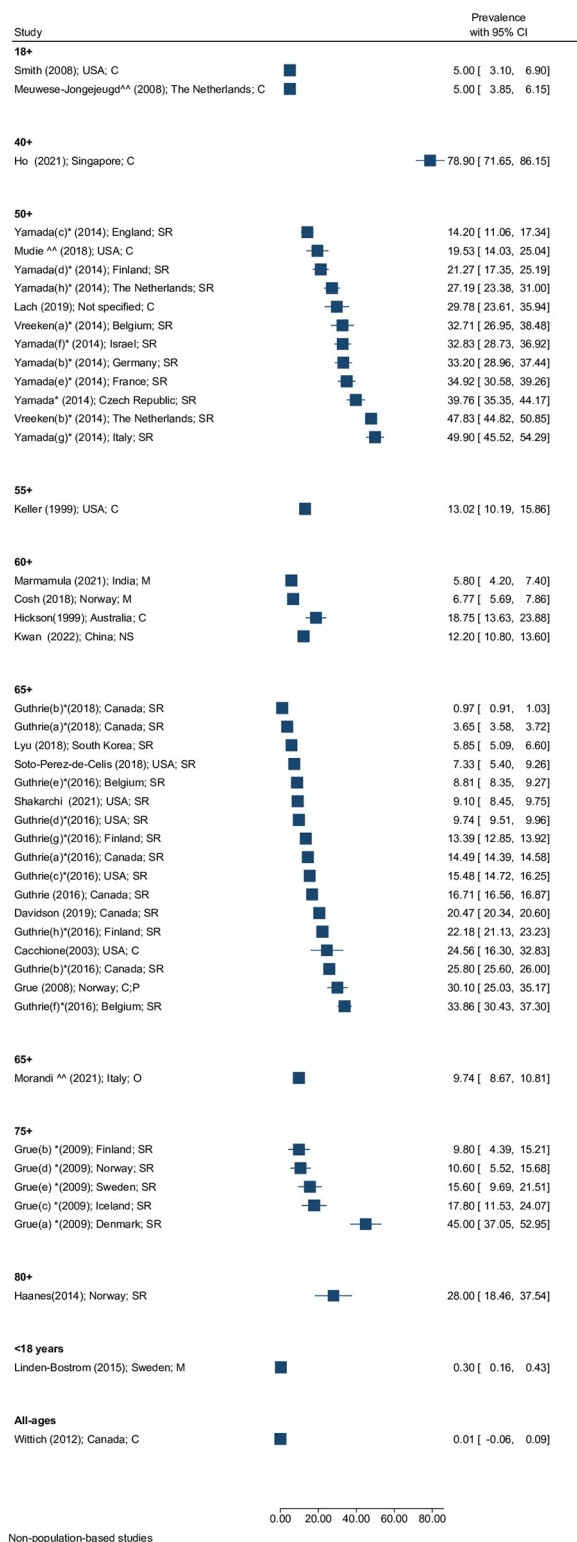

**Fig 3. Prevalence (95%CI) of DSI by age, region, study setting, and measurement type in non-population-based studies (SR = self-report; C = clinical (PTA or VA); M = mixed (SR and C); O = other)] ^^ measured DSI in a subpopulation, * same study, multiple countries, or multiple prevalence estimates reported.**

**Prevalence amongst adults ≥18 years.** Seven of ten population-based studies conducted amongst adults ≥18 years were in the USA, reporting a prevalence of DSI between 0.3% (n = 468,303; self-reported) [36] and 44.3% (n = 963; VA >0.3; PTA > 25 dB HL) [37]. Three other population-based studies were identified from Denmark [38], England [39] and Spain [40], with estimates of 0.02% (n = 10,000; self-report), 3.7% (n = 7,546; self-report), and 3.9% (n = 23,089; self-report) respectively. Two clinic-based reports, one in the USA and one in The Netherlands reported estimates of 5.0% (n = 400; VA <6/12 better eye; PTA ≥40dB better ear) and 5.0% (n = 1,359; VA>0.3; 3FA>25) respectively [41, 42].

**Prevalence amongst older adults.** Amongst older adults, prevalence was reported across ten different age categories. The estimated prevalence in population-based studies was highly variable across studies, from 0.97% in Japan (n = 2,190; self-report) [43], to 58.6% in China (10,575; self-report) [44]. Variation in prevalence was observed even when the same definitions were used. For example, in people 65 years and older using clinical assessment tools (VI threshold of <6/18 in better eye and HI threshold >25dB in the better ear); the prevalence of DSI ranged from 3.1% (USA; n = 446) to 21% (Australia; n = 1,611) [45–47]. Another example, amongst four reports using self-reported measures (Likert scale) with participants aged ≥50 years, the prevalence ranged between 3.8% (England; n = 4,621) [48] and 8.1% (USA; n = 13,092) [49]. Fig 2 provides details of the prevalence range of DSI found across different reports, by age group. S1 Table provides more details of the definitions used in each report.

In addition to the 121 reports of prevalence identified, three reviews also examined prevalence. Besser et al. (2018) found that the prevalence ranged between 3.1% in Australia and 18.2% in Japan [50]. Heine et al. (2015) reported that the prevalence across 42 included studies ranged between 3.3% and 64% and noted the range of criteria used [51]. Dewan et al. (2012) reported a lack of evidence on the prevalence of DSI due to congenital rubella syndrome [52].

## Impact of DSI

Of the 156 reports that measured at least one impact of DSI, 93 reported psychosocial outcomes, 55 reported participation outcomes, and 56 reported a physical health outcome. Further, nine reviews included outcomes related to psychosocial health, four included participation outcomes and two examined outcomes related to physical health. These three categories of outcomes are outlined further below.

Across the 156 reports of primary studies, 141 included a comparison group, though this group varied across studies (e.g. people without DSI, people with VI only, people with HI only, people without either sensory impairment). Over two thirds of the reports with a comparison group (n = 94, 67%) found worse outcomes for people with DSI, while a quarter (n = 34, 24%) reported varied outcomes (e.g. two outcomes worse among people with DSI, one the same) and a small number reported no difference between people with DSI and the comparator group (n = 13, 9%); no reports found better outcomes for people with DSI.

**Psychosocial health.** Of the 86 reports measuring psychosocial outcomes that included a comparator group, 72% (n = 62) showed worse outcomes in the DSI group in comparison to a control group while no difference was found between groups in 12% (n = 10) of reports (Fig 4a). Depression (n = 40) and cognition (n = 35) were the most reported psychosocial outcomes with a comparison group and people with DSI had worse outcomes in 68% of reports (n = 27) and 71% (n = 25) or reports respectively. All other outcomes are shown in Fig 4b and outlined in the S2 Table.

Of the seven reports without a comparator group, there was a trend towards negative outcomes for people with DSI. For example, Amini and colleagues found that war veterans in Iran with DSI had poor quality of life scores, measured using the SF-36 Health Survey [53]. In

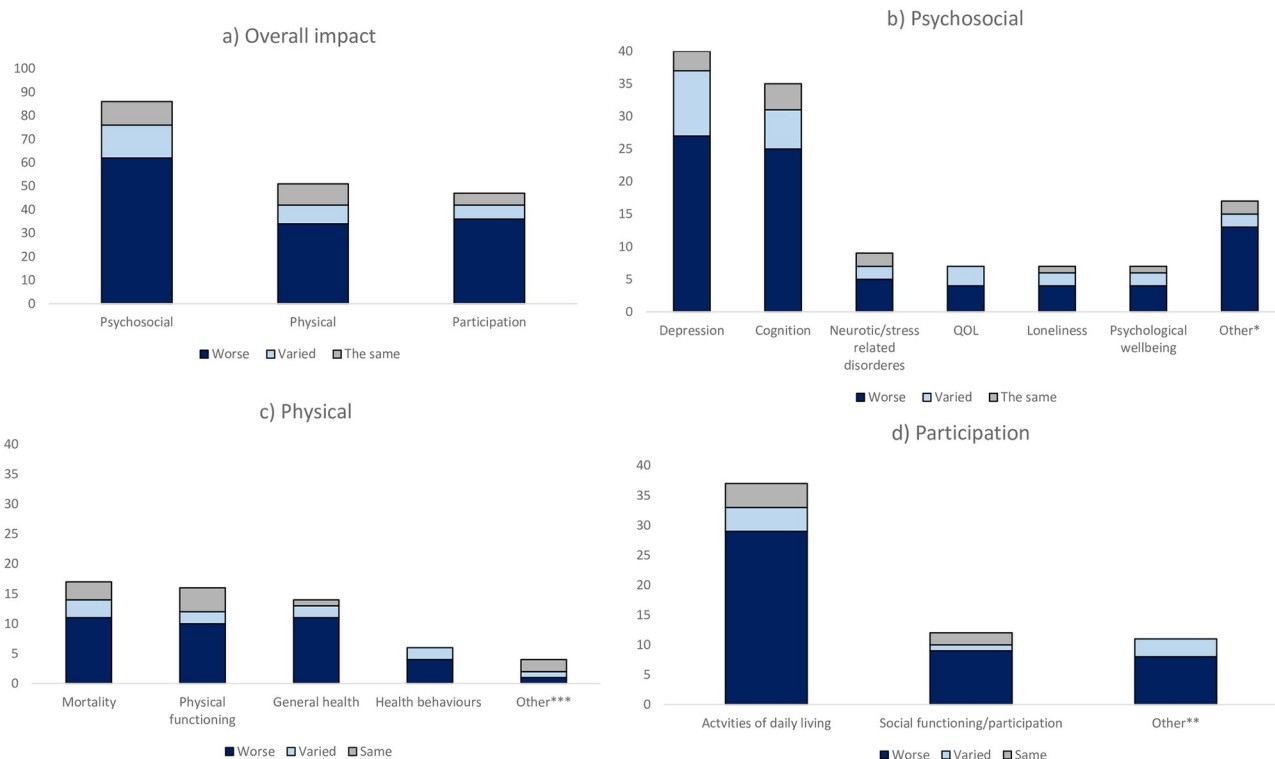

**Fig 4. Impact of DSI across the domains of psychosocial, participation and physical health outcomes (\*Other includes: Quality of life; self-evaluation of life; suicide ideation or attempt; episodic memory; behavioural disorder; developmental disability; acute confusion; intellectual disability.** \*\*Other includes: impendence, participation, retirement, education, wealth, self-regulation/goal pursuit, work, communication. \*\*\*Other includes: health care costs, sexual health, long term care admissions, hospitalisation)]. Studies were classified as "varied" if results for a study were worse in one outcome of interest or and the same or better in another.

another example, Appollonio showed that people with DSI had poor scores on self-evaluation of life, depression and general mental health [54]. In-depth interviews with people with DSI in several European countries described the stigma they experienced [55].

In addition to the primary reports in this category, nine reviews were identified that considered the psychosocial impacts of DSI. The majority of reviews reported that DSI is associated with poor psychological wellbeing–including depression [20, 22, 56], cognition [22, 51, 56], coping capacity [6], resilience [6], and quality of life [19]. DSI was also found to be associated with other disorders including autism [57]. One review found that those with Usher's syndrome, a major syndromic cause of DSI, were more at risk of developing psychological disorders, but there was no evidence that this was correlated with the presence of DSI [58]. Another review looked at psychosocial health among caregivers, but only one study was identified which showed no important impact of caregiving of people living with DSI on psychosocial health of the caregivers [59]. Several reviews reported methodological limitations with included studies, or insufficient information to draw strong conclusions.

**Physical health.** Of the 50 reports measuring and comparing physical health, over half found worse outcomes (n = 33, 66%), 16% (n = 8) found varied outcomes, and a similar proportion (n = 9, 18%) found no difference. Physical health outcomes were varied, and comparisons were most commonly reported for general health (n = 16), physical functioning (n = 16) and mortality (n = 16). General health was worse for people with DSI in 79% of reports (n = 11/14), mortality was higher in 69% (n = 11) of reports and physical functioning was worse in 63% of reports (n = 10).

The two reviews that summarised physical health outcomes reported that people with DSI had increased mortality [22, 51], poor health [51], and reduced functional status [51].

**Participation.** Of the 47 reports measuring and comparing participation-related outcomes, 77% (n = 36) found worse outcomes for the DSI group, 11% (n = 5) found both better and worse outcomes, and 13% (n = 6) found no difference (Fig 4a). Outcomes relating to participation were diverse and comparisons were most commonly reported for activities of daily living (n = 37) and social participation (n = 12) (Fig 4c). In studies reporting activities of daily living, people with DSI had worse outcomes 78% of reports (n = 29/37). Likewise, in studies reporting social participation in 75% of reports (n = 9/12) found worse outcomes in comparison to another group.

In general, the four reviews reporting participation outcomes suggested that people with DSI experience difficulties in participation in key areas of life, including communication [22, 23, 51], mobility [23], activities of daily living [23, 51], independence [6, 22], employment [51], social networks [6] and social interactions [23].

## Discussion

The purpose of this scoping review was to determine the nature and extent of the evidence on the prevalence and impact of DSI. Overall, 14 reviews and 183 reports of 153 unique primary studies were identified. Primary studies were mostly population-based (67%) and located in high-income countries (86%). We summarised prevalence and impact outcomes for a broad range of age groups and regions, which sets our review apart from previous reviews, which have focused on a particular age group or impact of DSI (e.g. quality of life).

Most of the research we identified was from North America or European countries, with almost no evidence from low- or middle-income countries (LMICs). There is a great need for more evidence to be generated from LMIC contexts. The magnitude and experiences of people with DSI in LMICs is likely to differ substantially to HICs. Given that the vast majority of HI and VI are experienced by people in LMICs [1, 2], there is potentially a substantial number of people experiencing DSI and its wider impacts. Evidence from LMICs on DSI could help to advocate and plan innovative service delivery models. In contexts with scarce human resources for both sensory impairments, integrated ear and eye care services may help to address the huge unmet need for diagnosis, treatment, rehabilitation, and policies for inclusive environments.

### Differences in definition

There was striking heterogeneity in the definitions used for DSI; with at least 75 alternative definitions used across the included reports. Despite the differences across studies, there were some commonalities. For example, the majority of studies included people with residual vision and/or hearing in their definitions, rather than only focusing on people with profound hearing loss or blindness. Broadly, DSI was measured using either self-report or clinical tools across studies, using a range of tools and thresholds.

Within the field of deafblindness, there are differing perspectives on whether it is best measured and defined using clinical tools (medical approach), or the resulting functional disability (functioning-based approach), or using the medical aetiology of DSI (e.g. CHARGE) [60]. Ask Larsen and colleagues discussed some of the reasons for different definitions in a review of deafblindness [60]. Another challenge with definitions of DSI, is that individuals cannot be easily categorised in to deafblind or not deafblind–the interaction between the impairments creates complexity that needs to be accommodated across a spectrum. Using definitions based on the medical approach will result in different numbers of people identified with DSI than if using functioning-based approach. This has been examined in the field of disability by

Mactaggart and colleagues, who found that using tools to assess functional limitation to measure the prevalence of disability would under-estimate the number of people with underlying clinical impairments [61]. The lack of consensus on a definition and assessment criteria for DSI makes it difficult to gather data that are comparable between studies, settings and over time. This is a key area for action to inform responsive services for people with DSI.

## Prevalence of DSI

The prevalence of DSI was reported for at least 15 different age groups, and a broad range of DSI definitions were used. The age included in the prevalence estimates has implications for service planning and delivery, which will be different for children, working age adults, and the elderly. Future studies should align age categories with those recommended by WHO (e.g., older adults are aged 60 years and older), and present data in a clear and standardised format [62, 63]. A number of studies in this review did not use population-based samples, instead reporting prevalence based on data from clinics, highlighting the lack of large-scale studies of DSI in the general population. The majority of the literature came from high-income countries, so any differences in prevalence rate between low- and high-income countries could not be examined in depth. This warrants further investigation, as the prevalence may be higher in LMICs due to factors such as poor access to services. Despite the lack of comparable data, the findings of our review suggest that the prevalence of DSI increases with age, with population-based studies of people aged ≥50 years reporting rates between 1.6% and 18.2% [48, 64–73]. The World Federation for the Deafblind report, which analysed 22 population-based surveys conducted in LMICs that measured DSI using self-report, found a pooled prevalence of 0.2%. Although our review only included 3 all age population-based studies, the prevalence in these ranged between 0.003% and 1.91% consistent with the low pooled prevalence found in the report [5]. The relatively high prevalence highlights that DSI should be considered in the delivery of stand-alone HI and VI services, particularly for older age groups. For example, providers could make accommodations for people with DSI within audiology clinics, via communicative support [56]. There is also an opportunity to screen for HI within vision services, or vice versa. Further, population-based surveys of single impairments (vision or hearing) should consider the potential overlap between the two conditions. In addition, the rehabilitation strategies that are suitable for people with single impairment may not be always suitable for people with DSI. For example, people with DSI may require support in the development of tactile communication techniques to supplement assistive devices (such as glasses or hearing aids) or visual sign language [56].

## Impact of DSI

Many of the identified reports examined the impact of DSI on people's physical and psychological health and on their ability to participate in key life areas. These provide, across a wide range of domains and indicators, a clear picture of the adverse impact of DSI, with two thirds reporting worse outcomes for all domains considered, and a further quarter reporting worse outcomes for some of the domains considered (varied outcome). The diverse range of indicators used across studies examining impact makes direct comparisons difficult, however the trends observed were clear—people with DSI may experience poorer general health, increased morbidity and mortality, and decreased participation in everyday activities. In particular, the review identified that people with DSI experienced more depression [45, 48, 74–89] and greater difficulties with performing activities of daily living [70, 76, 81, 83, 84, 87, 88, 90–111] than people without sensory impairments. In addition, there was an indication in a small number of studies that these outcomes were worse for people with DSI than for people with single impairments [74, 81, 87, 88, 97].

The findings of this scoping review concur with previous reviews focussing on either HI or VI [8–16]. For example, a recent review found hearing loss was associated with greater odds of depression in older adults across 35 studies [112]. Similarly, VI was found to be consistently associated with depression in a 2015 systematic review [113]. There is also evidence that HI [114] and VI are independently associated with increased mortality [115]. It is not clear from this scoping review whether the combined effects of both HI and VI result in increased risk of poorer health and wellbeing. This is an area that warrants further attention. The findings of grey literature sources such as the 2018 report by the World Federation of the Deafblind also agree with the findings of our review–people with deafblindness may have poorer levels of health, and poorer levels of participation in work and education [5].

Further, given the tendency for worse outcomes among people with DSI, healthcare professionals working in the field of eye care or hearing care have an opportunity to identify, and intervene early to help alleviate some of the negative consequences on physical and psychological health, as well as in participation in society. In particular, this review has identified a high need for mental health services for people with DSI. Evidence from high-income settings suggests that despite the high need for these services, there are substantial barriers to access, such as lack of qualified interpreters [116, 117].

## Limitations

Our findings must be interpreted in the context of several limitations. First, scoping reviews do not assess the risk of bias of included studies, thus the studies included in this review are likely to be of varying quality. Second, we excluded reports not in English, so we may have missed some evidence, including from LMICs. Third, by including review articles and primary reports in the review we may have introduced some duplication of studies; we attempted to counter this by presenting review findings separately to primary studies. Fourth, we only conducted electronic searches and did not handsearch journals. It is possible that we may have missed articles that were not properly indexed on the electronic databases, or where indexing was delayed–for example, a potentially relevant review on the prevalence of DSI [118] published in January 2022 was not indexed by the time of our searches in April 2022. However, the findings are consistent with the findings of our scoping review–that prevalence increases by age, and the findings across studies were often not comparable. Finally, when constructing the search strategy, we defined search terms for potential impact outcomes based on preliminary searches of the literature (e.g. mortality, independence, participation, vulnerability, quality of life, mental health) which may have led us to miss some studies reporting other impacts. We did, however, identify a wide variety across our three impact domains.

## Conclusion

This scoping review indicates that DSI is a relatively common condition, particularly among older adults. Moreover, the combination of HI and VI has a major impact on the physical, psychosocial, and participation experiences in the lives of affected people and is worthy of much more attention than it is currently receiving. In particular, people with DSI experience depression and decreased participation in everyday life. The magnitude of DSI is likely to increase with population ageing, and therefore research focused on this group is increasingly important. There is a gap in evidence from LMICs on the prevalence and impact of DSI. There is a pressing need for a consensus position on the definition(s) of DSI and standardisation of reporting age groups, to enable reliable estimates to be developed. Further there is an urgent need for research to identify the most effective strategies to improve access to health and wellbeing services for people with DSI. These findings are important for policy and practice when

trying to address the additional needs of people experiencing DSI, beyond the impairments alone.

## Supporting information

**S1 Checklist. Preferred Reporting Items for Systematic reviews and Meta-Analyses extension for Scoping Reviews (PRISMA-ScR) checklist.**
(DOCX)

**S1 Text. Search strategy.**
(DOCX)

**S1 Table. Studies reporting prevalence of dual sensory impairment (DSI) by age group.**
(DOCX)

**S2 Table. Reports measuring psychosocial health outcomes for people with dual sensory impairment (DSI).**
(DOCX)

**S3 Table. Reports measuring participation outcomes for people with dual sensory impairment (DSI).**
(DOCX)

**S4 Table. Reports measuring physical health outcomes.**
(DOCX)

**S5 Table. Included systematic reviews.**
(DOCX)

**S6 Table. Included studies (n = 197).**
(DOCX)

## Author Contributions

**Conceptualization:** Tess Bright, Jacqueline Ramke, Bonnielin K. Swenor, Matthew J. Burton, Jennifer R. Evans.

**Data curation:** Tess Bright, Jacqueline Ramke, Justine H. Zhang, Gatera Fiston Kitema, Sare Safi, Shaffi Mdala, Miho Yoshizaki, Christopher G. Brennan-Jones, Islay Mactaggart, Iris Gordon, Jennifer R. Evans.

**Formal analysis:** Tess Bright, Justine H. Zhang, Gatera Fiston Kitema, Sare Safi, Shaffi Mdala, Miho Yoshizaki, Christopher G. Brennan-Jones, Islay Mactaggart, Jennifer R. Evans.

**Funding acquisition:** Matthew J. Burton.

**Investigation:** Justine H. Zhang, Gatera Fiston Kitema, Sare Safi, Shaffi Mdala, Miho Yoshizaki, Christopher G. Brennan-Jones, Islay Mactaggart, Jennifer R. Evans.

**Methodology:** Tess Bright, Jacqueline Ramke, Jennifer R. Evans.

**Supervision:** Jacqueline Ramke, Matthew J. Burton, Jennifer R. Evans.

**Writing – original draft:** Tess Bright.

**Writing – review & editing:** Tess Bright, Jacqueline Ramke, Justine H. Zhang, Gatera Fiston Kitema, Sare Safi, Shaffi Mdala, Miho Yoshizaki, Christopher G. Brennan-Jones, Islay Mactaggart, Iris Gordon, Bonnielin K. Swenor, Matthew J. Burton, Jennifer R. Evans.

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
