## [Decision Letter · Decision Letter 0]

13 Feb 2023

PGPH-D-22-01983

Prevalence and impact of combined vision and hearing (dual sensory) impairment: a scoping review

Dear Dr. Bright,

Thank you for submitting your manuscript to PLOS Global Public Health. After careful consideration, we feel that it has merit but does not fully meet PLOS Global Public Health’s publication criteria as it currently stands. Therefore, we invite you to submit a revised version of the manuscript that addresses the points raised during the review process.

Your manuscript has been evaluated by two reviewers, and their comments are available below and in the attached document.

The reviewers have raised a number of concerns that need attention. Both reviewers have made requests for clarification and further discussion, but please note in particular the comments from Reviewer #2 regarding details in the reporting of the methodology, as well as clarity and consistency in the Results section. Please ensure you address each of the reviewers' comments in full when revising your manuscript.

We look forward to receiving your revised manuscript.

Kind regards,

Hugh Cowley

Staff Editor

Journal Requirements:

1. Please ensure that the CRediT author contributions listed for every co-author are completed accurately and in full. 

At this stage, the following Authors require contributions: All Authors. Please ensure that the full contributions of each author are acknowledged in the ""Add/Edit/Remove Authors"" section of our submission form.

The list of CRediT author contributions may be found here: 

https://journals.plos.org/globalpublichealth/s/authorship#loc-author-contributions 

2. Please change article type to research article, not review.

3. Please amend your detailed Financial Disclosure statement. This is published with the article. It must therefore be completed in full sentences and contain the exact wording you wish to be published.

a. State what role the funders took in the study. If the funders had no role in your study, please state: “The funders had no role in study design, data collection and analysis, decision to publish, or preparation of the manuscript.”

4. Please send a completed 'Competing Interests' statement, including any COIs declared by your co-authors. If you have no competing interests to declare, please state "The authors have declared that no competing interests exist". 

5.In the online submission form you indicate that your data is not available for proprietary reasons and have provided a contact point for accessing this data. Please note that your current contact point is a co-author on this manuscript. According to our Data Policy, the contact point must not be an author on the manuscript and must be a third party. Please revise your data statement to a non-author institutional point of contact, such as a data access or ethics committee, and send this to us via return email. Please also include contact information for the third party organization, and please include the full citation of where the data can be found.

6. Please provide separate figure files in .tif or .eps format and remove the embedded figures from the manuscript file.

7. We have noticed that you have uploaded Supporting Information files, but you have not included a list of legends. Please add a full list of legends for your Supporting Information files after the references list. 

Additional Editor Comments (if provided):

Reviewers' comments:

Reviewer's Responses to Questions

**Comments to the Author**

1. Does this manuscript meet PLOS Global Public Health’s publication criteria? Is the manuscript technically sound, and do the data support the conclusions? The manuscript must describe methodologically and ethically rigorous research with conclusions that are appropriately drawn based on the data presented.

Reviewer #1: Yes

Reviewer #2: Yes

2. Has the statistical analysis been performed appropriately and rigorously?

Reviewer #1: Yes

Reviewer #2: Yes

3. Have the authors made all data underlying the findings in their manuscript fully available (please refer to the Data Availability Statement at the start of the manuscript PDF file)?

Reviewer #1: Yes

Reviewer #2: Yes

4. Is the manuscript presented in an intelligible fashion and written in standard English?

Reviewer #1: Yes

Reviewer #2: Yes

5. Review Comments to the Author

Reviewer #1: Please see attached document for detailed feedback. Overall, great work with the paper. There are some recommendations provided to ensure the quality of the paper is improved and greater clarity for readers. Minor revisions are needed.

Reviewer #2: Thank you for giving me the opportunity to review this excellent scoping review. I need to declare right at the beginning that this manuscript covers a topic that the field of deafblindness has been waiting for, and its content will likely make it a highly cited contribution to the literature. I am most excited that this work has been done and I congratulate the authors on their quality.

Having said that, there are a few minor items that I believe are important to be incorporated, to give the work a touch more subtlety and context. I provide these here in the spirit of elevating an already excellent manuscript even further:

The phrase “there is currently no consensus on how to define DSI,…” is not entirely correct, given that the Nordic definition of deafblindness is widely used and accepted in the field of deafblindness rehabilitation and service delivery: Centre for Welfare and Social Issues. (2016). The Nordic definition of deafblindness. http://www.kuurosokeat.fi/tiedosto/nordic_definition.pdf

In addition, there are definitions that follow the medical model in the context of eligibility criteria for services (e.g., the authors cite Wittich et al. 2012 which contains an eligibility grid for service access). It is however true that there is disagreement between whether a standard definition should rely on behaviorally measurable observations (e.g., acuity, field, audiogram) or functional impairment (subjectively reported). This disagreement is not well represented in the introduction at present.

As the authors mention, the variability in all aspects is remarkable, and problematic. I believe that the authors should use this variability to justify at the beginning of the methods section why a scoping review is indeed the correct choice of methodology, over other approaches to reviewing the literature.

Given the variability and sparsity of information about deafblindness prevalence, many previous studies have had to rely on government sources for data on the topic. Methodologically here, it is solid to rely on the peer-reviewed literature, but given the history and young age of the field of deafblindness, I suggest that the authors include a short comparison of their findings with some of the sources that are widely known to researchers in deafblindness, such as :

European Deafblind Network. (2014). European Deafblind Indicators: Mapping opportunities for deafblind people across Europe.

United Nations Department of Economic and Social Affairs Population Division. (2022). World Population Prospects 2022: Summary of Results - DESA/POP/2022/TR/NO. 3. Author. https://www.un.org/development/desa/pd/sites/www.un.org.development.desa.pd/files/wpp2022_summary_of_results.pdf

Robertson, J., & Emerson, E. (2010). Estimating the number of people with co-occurring vision and hearing impairments in the UK.

World Federation of the Deafblind. (2018). At Risk of Exclusion from CRPD and SDGs Implementation: Inequality and Persons with Deafblindness. Initial Global Report 2018. https://wfdb.eu/wfdb-report-2018/

The authors need to acknowledge somewhere that individuals cannot be simply categorized into deafblind or not deafblind. Part of the confusion about criteria is that this condition comes on a sliding scale and the interaction between the two impairment creates complexity that needs to be accommodated across a spectrum.

Line 298 “…but there was no evidence that this was correlated with DSI itself.” Do you mean correlated with level of severity or with the presence of DSI – here is one of those moments where that subtly of the spectrum may matter in the interpretation.

If the authors want to incorporate possibilities for improvement in the future, one example is the presentation of the data that are by the Global Burden of Disease team on rehabilitation:

Cieza, A., Causey, K., Kamenov, K., Hanson, S. W., Chatterji, S., Vos, T., Bill, F., & Foundation, M. G. (2020). Global estimates of the need for rehabilitation based on the Global Burden of Disease study 2019 : a systematic analysis for the Global Burden of Disease Study 2019. The Lancet, 6736(20), 1–12. https://doi.org/10.1016/S0140-6736(20)32340-0

Both vision and hearing are reported, but not their overlap. This needs to be addressed in the future, and the current review should hint at this gap in the methodology.

6. PLOS authors have the option to publish the peer review history of their article (what does this mean?). If published, this will include your full peer review and any attached files.

**Do you want your identity to be public for this peer review?** For information about this choice, including consent withdrawal, please see our Privacy Policy.

Reviewer #1: **Yes: **Abinethaa Paramasivam

Reviewer #2: No

---

## [Decision Letter · Decision Letter 1]

18 Apr 2023

Prevalence and impact of combined vision and hearing (dual sensory) impairment: a scoping review

PGPH-D-22-01983R1

Dear Dr Bright,

We are pleased to inform you that your manuscript 'Prevalence and impact of combined vision and hearing (dual sensory) impairment: a scoping review' has been provisionally accepted for publication in PLOS Global Public Health.

Best regards,

Julia Robinson

Executive Editor

Reviewer Comments (if any, and for reference):

Reviewer's Responses to Questions

**Comments to the Author**

1. If the authors have adequately addressed your comments raised in a previous round of review and you feel that this manuscript is now acceptable for publication, you may indicate that here to bypass the “Comments to the Author” section, enter your conflict of interest statement in the “Confidential to Editor” section, and submit your "Accept" recommendation.

Reviewer #1: All comments have been addressed

Reviewer #2: All comments have been addressed

2. Does this manuscript meet PLOS Global Public Health’s publication criteria? Is the manuscript technically sound, and do the data support the conclusions? The manuscript must describe methodologically and ethically rigorous research with conclusions that are appropriately drawn based on the data presented.

Reviewer #1: Yes

Reviewer #2: Yes

3. Has the statistical analysis been performed appropriately and rigorously?

Reviewer #1: Yes

Reviewer #2: Yes

4. Have the authors made all data underlying the findings in their manuscript fully available (please refer to the Data Availability Statement at the start of the manuscript PDF file)?

Reviewer #1: Yes

Reviewer #2: Yes

5. Is the manuscript presented in an intelligible fashion and written in standard English?

Reviewer #1: Yes

Reviewer #2: Yes

6. Review Comments to the Author

Reviewer #1: Thank you for addressing the suggestions provided and for providing your input for each comment. This is a very important contribution to the field. Thank you for your hard work with this research.

Reviewer #2: Congratulations on a revision well done!

7. PLOS authors have the option to publish the peer review history of their article (what does this mean?). If published, this will include your full peer review and any attached files.

**Do you want your identity to be public for this peer review?** For information about this choice, including consent withdrawal, please see our Privacy Policy.

Reviewer #1: **Yes: **Abinethaa Paramasivam

Reviewer #2: **Yes: **Walter Wittich
